# Graphene-Based TiO₂ Nanocomposite for Photocatalytic Degradation of Dyes in Aqueous Solution under Solar-Like Radiation

**Martina Kocijan** [1,*], **Lidija Ćurković** [1,*], **Davor Ljubas** [1], **Katarina Mužina** [2], **Ivana Bačić** [3], **Tina Radošević** [4], **Matejka Podlogar** [4], **Igor Bdikin** [5], **Gonzalo Otero-Irurueta** [5], **María J. Hortigüela** [5] and **Gil Gonçalves** [5,*]

[1]  Faculty of Mechanical Engineering and Naval Architecture, University of Zagreb, Ivana Lučića 5, 10000 Zagreb, Croatia; davor.ljubas@fsb.hr
[2]  Faculty of Chemical Engineering and Technology, University of Zagreb, Trg Marka Marulića 20, 10000 Zagreb, Croatia; kmuzina@fkit.hr
[3]  Forensic Science Centre "Ivan Vučetić", Ministry of the Interior, Ilica 335, 10000 Zagreb, Croatia; ivana.bacic@mup.hr
[4]  Jožef Stefan Institute, Jamova Cesta 39, SI-1000 Ljubljana, Slovenia; tina.radosevic@ijs.si (T.R.); matejka.podlogar@ijs.si (M.P.)
[5]  Centre for Mechanical Technology and Automation (TEMA), Mechanical Engineering Department, University of Aveiro, 3810-193 Aveiro, Portugal; bdikin@ua.pt (I.B.); otero.gonzalo@ua.pt (G.O.-I.); mhortiguela@ua.pt (M.J.H.)
\*  Correspondence: martina.kocijan@fsb.hr (M.K.); lidija.curkovic@fsb.hr (L.Ć.); ggoncalves@ua.pt (G.G.)

**Abstract:** This study presents a novel method for the development of TiO₂/reduced graphene oxide (rGO) nanocomposites for photocatalytic degradation of dyes in an aqueous solution. The synergistic integration of rGO and TiO₂, through the formation of Ti–O–C bonds, offers an interesting opportunity to design photocatalyst nanocomposite materials with the maximum absorption shift to the visible region of the spectra, where photodegradation can be activated not only with UV but also with the visible part of natural solar irradiation. TiO₂@rGO nanocomposites with different content of rGO have been self-assembled by the hydrothermal method followed by calcination treatment. The morphological and structural analysis of the synthesized photocatalysts was performed by FTIR, XRD, XPS, UV-Vis DRS, SEM/EDX, and Raman spectroscopy. The effectiveness of the synthesized nanocomposites as photocatalysts was examined through the photodegradation of methylene blue (MB) and rhodamine B (RhB) dye under artificial solar-like radiation. The influence of rGO concentration (5 and 15 wt.%) on TiO₂ performance for photodegradation of the different dyes was monitored by UV-Vis spectroscopy. The obtained results showed that the synthesized TiO₂@rGO nanocomposites significantly increased the decomposition of RhB and MB compared to the synthesized TiO₂ photocatalyst. Furthermore, TiO₂@rGO nanocomposite with high contents of rGO (15 wt.%) presented an improved performance in photodegradation of MB (98.1%) and RhB (99.8%) after 120 min of exposition to solar-like radiation. These results could be mainly attributed to the decrease of the bandgap of synthesized TiO₂@rGO nanocomposites with the increased contents of rGO. Energy gap ($E_g$) values of nanocomposites are 2.71 eV and 3.03 eV, when pure TiO₂ particles have 3.15 eV. These results show the potential of graphene-based TiO₂ nanocomposite to be explored as a highly efficient solar light-driven photocatalyst for water purification.

**Keywords:** graphene oxide; TiO₂@rGO nanocomposite; photocatalysis; solar-like irradiation; Methylene blue; Rhodamine B

## 1. Introduction

Nowadays, accessible clean water and energy resources are among the highest priorities for sustainable economic growth and societal wellbeing. Water scarcity is a growing worldwide problem, so it is imperative that wastewater is treated and re-used in industrial

processes, irrigation, and feeding to livestock, thus saving potable water for drinking. Research activities focus on the synthesis of new materials for use in energy-efficient processes for water treatment by using doped metal oxides with a reduced bandgap, which allows photocatalysis in the visible region of the electromagnetic spectrum [1,2]. Different types of pollutants such as pharmaceuticals, dyes, pesticides, etc., are widespread and may interact with the environment through different pathways. In the present work, as relevant environmental pollutants, methylene blue (MB) dye and rhodamine B (RhB) dye were selected as complex compounds for decomposition from an aqueous medium because of their often-frequent occurrence in the environment. MB and RhB are widely utilized for industrial production purposes such as dyeing and printing on textile, leathers, papers, and also on plastics [3–5].

Organic pollutants are entering the water cycle through wastewater release into the environment. Very toxic and cancerogenic residual dyes and their metabolites already have a strong impact on ecosystems. Moreover, these organic pollutants can be very harmful to living organisms, especially if they enter the food chain [6–10]. Thus, the removal of them from water is a huge challenge and an emergency assignment. Conventional wastewater treatment plants (WWTP) are not adequately designed for the removal of dyes, their metabolites, or transformation products. Hence, the removal of dyes and their degradation products from industrial wastewater before discharging into the environment requires necessary treatment [11,12].

To degrade the mentioned harmful compounds present in the environment, researchers are globally looking for an effective approach such as Advanced oxidation processes (AOPs). Nowadays, the AOPs have received considerable attention as alternative wastewater treatment processes, which may be implemented in conventional wastewater treatment processes to ameliorate their removal efficiency. The photocatalytic oxidation process, especially heterogeneous photocatalysis, is one of the AOPs that is increasingly being explored due to its relatively efficient degradation and environmentally benign impact [6,13–16]. This technology can be a clean, green, and sustainable alternative to conventional wastewater treatment technologies [17,18]. Currently, there are different materials that can be used as heterogeneous photocatalysts, but titanium dioxide ($TiO_2$) arises like the one with widespread use. This semiconductor material presents extraordinary properties for degradation of organic matter dissolved in water; its low environmental impact, cheapness, high chemical and thermal stability, and pollution-free process to remove low concentration pollutants, as well as biocompatibility, are relevant features for its application [2,19–21]. $TiO_2$ photocatalysts are usually very active under ultraviolet (UV) irradiation when the wavelength is lower than 387 nm. The major pitfalls in the practical application of heterogeneous $TiO_2$ based semiconductor photocatalysis are the wide bandgap energy and quick recombination of the photogenerated electron-hole pairs [22,23]. Therefore, the application of $TiO_2$ as a photocatalyst shows low efficiencies in the decomposition of effluent organic compounds under solar light. In global solar spectra that reach the Earth's surface, UV part amounts around 3–5%, while almost 45% is visible light [9,11]. New photocatalysts are being designed in the form of nanocomposite materials to overcome their current limitations assigned to the fact of being mostly active only under UV radiation [22–25].

Graphene-based materials have unique properties that may be applied in wastewater purification. Graphene oxide (GO) was chosen to prepare the graphene-based nanocomposites due to its affordable production of bulk quantities. The oxidation process of graphite to GO brings defects into its structure that change its physico-chemical properties [26]. As a result, the carbon network of GO possesses a substantial density of oxygen functional groups like epoxy, carbonyl, hydroxyl, and carboxyl groups. The oxygen functionalities may be removed by the reduction process, which can significantly shift its properties, such as improving optical surfaces and changing the electronic structure. The properties of reduced graphene oxide (rGO) play an important role in the synthesis of graphene-based

nanocomposite materials. Moreover, the improved performance of developed composites will rely on the influence of rGO and materials committed to its surface [26–28].

In this study, the synthesis parameters of the one-pot cost-effective hydrothermal procedure were optimized, followed by a calcination treatment at 300 °C to prepare $TiO_2$@rGO nanocomposite-based photocatalyst. As model pollutants, MB and RhB dyes were exposed to the photocatalytic decomposition initiated by the solar-like irradiation source. The aim of this research was to contribute to a broader understanding of photocatalytic degradation of MB and RhB with additional criteria to prepare $TiO_2$@rGO nanocomposite, showing significant photocatalytic activity.

## 2. Materials and Methods

### 2.1. Chemicals and Reagents

Commercially available flakes of graphite (particle size $\leq 50$ μm) and titanium (IV) isopropoxide ($Ti(C_3H_5O_{12})_4$, TTIP 97%) were acquired from Sigma-Aldrich (St. Louis, MO, USA), while nitric acid ($HNO_3$, $\geq 65\%$) and sulfuric acid ($H_2SO_4$, 97%) were obtained from Honeywel Fluka (Seelze, GER). Hydrochloric acid (HCl, 37%), potassium permanganate ($KMnO_4$) and sodium nitrate ($NaNO_3$), hydrogen peroxide ($H_2O_2$, 30% $w/v$), i-propanol ($C_3H_7OH$), acetylacetone ($CH_3(CO)CH_2(CO)CH_3$) were supplied from Gram mol (Zagreb, CRO). Methylene blue (MB), (7-(dimethylamino) phenothiazine 3-ylidene)-dimethyl azanium chloride and Rhodamine B (RhB), (9-(2-carboxyphenyl)-6-diethylamino-3-xanthenylidene)-diethylammonium chloride were purchased from Sigma-Aldrich (St. Louis, MO, USA) The solutions with stated chemicals were prepared using ultrapure water (Millipore).

### 2.2. Synthesis of Graphene Oxide (GO)

A chemical process called Hummer's method was used to manufacture the GO [29]. Firstly, flakes of graphite powder (3 g) were dispersed in a concentrated solution of $H_2SO_4$ (69 mL) then $NaNO_3$ (1.5 g) was added to an Erlenmeyer flask. The solution was stirred on a magnetic stirrer until a homogeneous solution was obtained at a low temperature (0–5 °C). Then $KMnO_4$ (9.0 g) was gradually added to keep the reaction temperature below 20 °C. The mixture was warmed to 35 °C and shuffled for 30 min, during which 138 mL of water was slowly added. The mixture produced an exothermic reaction, spontaneously heating to 98 °C. During the next 15 min, the mixture was heated in order to hold the temperature at 98 °C, and then the reaction mixture was chilled in a water container for a few minutes to the room temperature. Then 420 mL of water and 3 mL of $H_2O_2$ (30%) were added to the mixture. The resultant suspension was centrifuged at 3000 rpm for 10 min to remove the remaining impurities and to recover the GO. The isolated GO was first washed with HCl (10%) and then rinsed intensively with deionized water until it reached neutral pH.

### 2.3. Preparation of $TiO_2$ Colloidal Solution

The main precursor of titanium, TTIP, was added drop by drop in the *i*-propanol. Acetylacetone chelating agent was then added to the solution, followed by nitric acid acting as a catalyst. The chemicals were mixed in the molar ratio, 1:35:0.63:0.015. During the preparation, the mixture solution was constantly stirred, and clear yellow color of solvent was attained [30].

### 2.4. Preparation of $TiO_2$@rGO Nanocomposites

The $TiO_2$ sol and GO nanosheets were mixed to produce $TiO_2$@rGO nanocomposites by hydrothermal method. Different ratios of the prepared suspension of GO were added into a colloidal solution of $TiO_2$ sol and stirred for 1 h. After that, the black-brown mixture was homogenized for 10 min in an ultrasonic bath. The obtained product was transferred into a Teflon-lined autoclave tube. The hydrothermal reaction was performed at 180 °C for 8 h. The resultant nanocomposite material was extensible washed first with *i*-propanol and

then with deionized water until the neutral pH value was reached. The obtained $TiO_2$@rGO nanocomposite was dried at 60 °C in the electric dryer for 1 h and then calcinated in the oven at 300 °C for 1 h.

### 2.5. Characterization of Photocatalysts

The crystalline phases of GO and $TiO_2$@rGO were analyzed by powder X-ray diffraction (P-XRD, Rigaku) operating with CuK$\alpha$ radiation. Fourier transform infrared (FT-IR/ATR, Bruker Vertex 70)) spectra were performed to analyze the chemical groups in the structure of the nanocomposite materials obtained. To confirm the crystalline phase of GO and $TiO_2$@rGO, Micro-Raman analyses were performed using a Bruker SENTERRA spectrometer with an Olympus microscope. X-ray photoelectron spectroscopy (XPS) performed on SPECS Phoibos 150 with AlK$\alpha$ radiation (1486.74 eV) was used for elemental composition determination. Binding Energy (BE) was corrected using the main peak of reduced GO as a reference, set at 284.4 eV. C 1s spectra of GO and rGO were normalized to get similar intensity to the other carbon spectra. Scanning electron microscopy and energy-dispersive X-ray spectroscopy (SEM/EDX) were carried out with the aim of obtaining microstructural characteristics and elemental mapping of the samples. Measurements were made using Hitachi TM4000 plus tabletop SEM-EDX facility with 15 kV field energy in the backscattered electron (BSE) imaging mode and FEG-SEM model JSM-7600F from Jeol at 10 kV in the secondary electron (SE) mode. The bandgap energy of the prepared nanocomposite powders was acquired from UV/Vis/NIR spectroscopy measurements using a Perkin Elmer Lambda 950 spectrophotometer.

### 2.6. Photodegradation Experiments

Methylene blue dye (MB) and Rhodamin B dye (RhB) purchased from Sigma-Aldrich (St. Louis. MO, USA) have been used for monitoring the photoactivity of the synthesized $TiO_2$@rGO nanocomposites. Each pollutant was dissolved in MilliQ water in order to reach a concentration of 10 mg/L. The photocatalysts (4.5 mg) were dispersed in 9 mL of the contaminant solution within a modified beaker reactor with quartz cover. As-prepared samples were left for 60 min in a chamber without irradiation. After the irradiation was turned on, the photocatalytic efficiency was monitored over 120 min. The magnetic stirrer was continuously turned on. Concentration change was measured by the adsorption decrease using a UV/VIS spectrophotometer (PerkinElmer Lambda 950) in set time intervals while irradiated by light in a climatic chamber equipped with the cooling system, using Osram's Ultra Vitalux lamp. The lamp produces a mix of radiation intervals that can be found in the natural solar radiation spectrum. Therefore, the term solar-like radiation for this radiation mix was used. The lamp and the solution in the reactor were set up at a distance of 20 cm.

From the slope of the straight line, the first-order rate constant was calculated, represented with the following equation [31]:

$$A_t = A_0 \cdot e^{-k \cdot t} \tag{1}$$

where $k$ (min$^{-1}$) is the rate constant of photodecomposition of dye, $A$ is the absorption of dye at the time of the photocatalytic process, and $A_0$ is the absorption at the beginning of the experiment [32,33].

The calculation of half-life ($t_{1/2}$) was done using the following equation [31]:

$$t_{1/2} = \frac{ln2}{k} \tag{2}$$

The percentage of photocatalytic degradation efficiency of the prepared photocatalysts were calculated using the following equation [31]:

$$\eta = \frac{A_0 - A_t}{A_0} \cdot 100\% \tag{3}$$

where $\eta$ is the percentage efficiency of the photodecomposition of dye, $A_0$ is the absorbance of the starting content of dye, and $A_t$ is the content of dye after irradiation at the time, $t$ (min) during the photocatalytic experiment.

## 3. Results and Discussion

### 3.1. Characterization of Photocatalyst

The successful preparation of materials by the combination of the hydrothermal method and calcination can be demonstrated with detailed characterization of the obtained TiO$_2$@rGO nanomaterials. The FT-IR spectra of natural graphite GO and rGO are presented in Figure 1A, while the spectra of TiO$_2$ and the nanocomposites are shown in Figure 1B. The GO spectrum present the following peaks at 3372 cm$^{-1}$ corresponded to the stretching vibration of the carboxyl group (−OH), which was ascribed on the attendance of alcohol groups and absorbed water molecules, 2857 cm$^{-1}$ and 2925 cm$^{-1}$ vibrations corresponded to the symmetric and asymmetric CH$_2$ stretching of GO [34,35]. The stretching vibration at 1715 cm$^{-1}$ ascribed to the C=O from carbonyl and carboxyl group and the vibration at 1622 cm$^1$ (C=C) as skeletal vibration of unoxidized graphitic materials [35]. The deformation vibration of C–OH stretch from the alcohol group was displayed at 1376 cm$^{-1}$ [36,37]. The stretching vibrations at 1221 cm$^{-1}$ were assigned to the C–O–C of epoxy groups, and at 1039 cm$^{-1}$ were ascribed to the C–O from the alkoxy group [37]. By comparing the GO and rGO spectra, displayed in Figure 1A, it could be seen that in the rGO sample, that the intensity of peaks attributed to oxygen functional groups, e.g., O–H, C=O, and C–O, were reduced [38]. The TiO$_2$@rGO nanocomposite spectra exhibited peaks between 450–900 cm$^{-1}$, which belonged to the stretching vibration of Ti–O–Ti and Ti–O–C bonds, affirming the effective interaction between Ti and C [39,40].

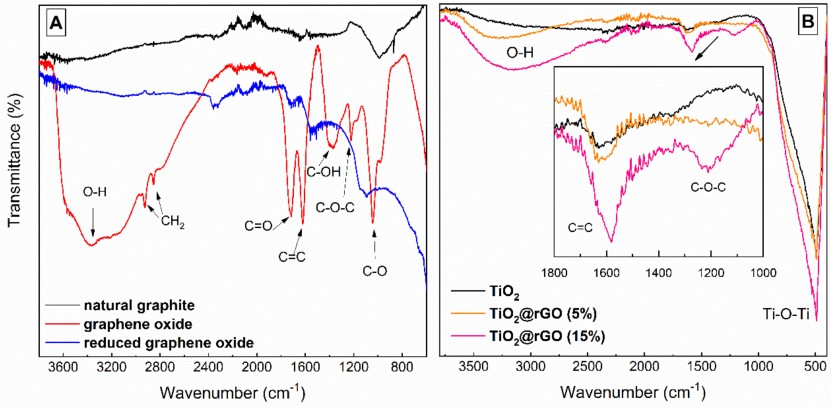

**Figure 1.** FTIR spectra of (**A**) natural graphite, GO, rGO; (**B**) TiO$_2$ and TiO$_2$@rGO nanocomposites prepared with different amounts of GO (5 wt.% and 15 wt.%).

The Raman spectrum of TiO$_2$ is shown in Figure 2A (inset), highlighting five sharp peaks for the anatase-phase of TiO$_2$. These peaks are located at 147 cm$^{-1}$, 198 cm$^{-1}$, and 641 cm$^{-1}$ as Eg, 399 cm$^{-1}$ as B1g, and 517 cm$^{-1}$ as A1g + B1g [41]. As the figures show, these peaks appeared for all prepared nanocomposites, but in comparison to pure TiO$_2$ nanoparticles, the intensity of the anatase phase was strongly reduced in the nanocomposites. Figure 2A shows, the prepared nanocomposites with a smaller amount (5 wt.%) of rGO had sharper peaks than the prepared nanocomposites with a higher amount (15 wt.%). The crystalline phase of the nanocomposite was disturbed because of the content of rGO. Figure 2B presents the Raman spectra of GO, rGO, and nanocomposites. The GO spectrum showed the presence of the two characteristic peaks at 1329 cm$^{-1}$ as D-band and 1585 cm$^{-1}$ as G-band. On the rGO spectrum, two peaks appeared at 1347 cm$^{-1}$ (D-band) and 1593 cm$^{-1}$ (G-band). The vibrations of the peaks in rGO materials were shifted significantly in relation to GO materials. The reason for the shift is the structure of the graphene material, where the G band corresponds to the sp$^2$ hybridization in C-C bonds, and D band

matches the sp$^3$ defects in carbon atoms as associated with structural defects [42,43]. The spectra of the prepared nanocomposites show characteristic peaks of anatase-phase TiO$_2$ and the presence of the D and G band. The binding of TiO$_2$ nanoparticles into rGO, as well as the intensities of anatase-phases in the nanocomposites, showed a significant decrease (Figure 2A,B).

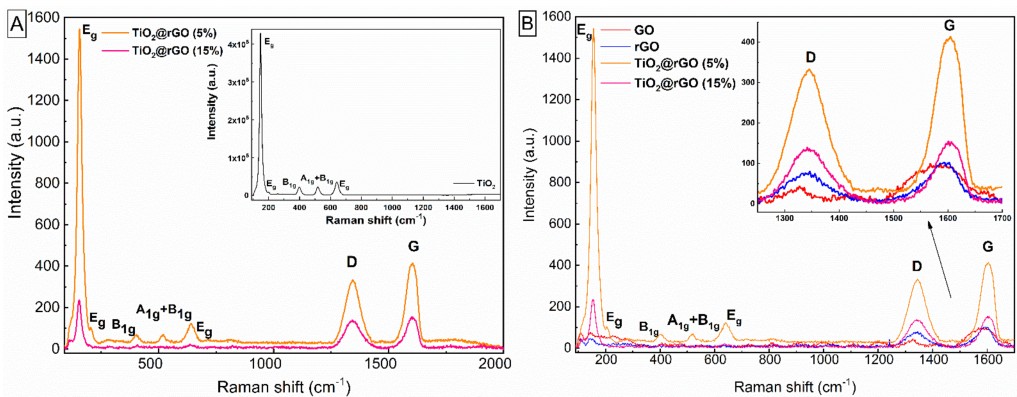

**Figure 2.** Raman spectra of (**A**) the respective nanocomposites, (inset shows TiO$_2$ calcinated at 300 °C); and (**B**) GO, rGO, and the respective nanocomposites. The inset shows the characteristic D and G bands of graphene-based materials.

As Figure 2B shows and Table 1 confirms, the content of graphene has an influence on the intensity of G and D bands in the nanocomposites. The reason for the high-intensity ratio between D and G bands was the presence of a lot of sp$^2$ hybridization in C-C bonds [41]. The rGO shows a higher ratio than GO, indicating that the rGO contained more defects. The nanocomposites have a higher intensity ratio compared to GO indicating higher graphene structural disorder upon binding onto TiO$_2$ [43]. The interactions of Ti-O-C bands could be the reason for increasing sp$^3$ defects in nanocomposites [42]. The intensity ratios between D and G bands were utilized to determine the crystal size parallel to basal planes ($L_a$), using the equation of Tuinstra, where the coefficient 38.5 was for wavelength at 633 nm [44]:

$$L_a(nm) = \frac{38.5}{I_D/I_G} \tag{4}$$

**Table 1.** Raman intensity and shift of the prepared GO, rGO, and TiO$_2$@rGO nanocomposites.

| Samples ID | $D$-Band, cm$^{-1}$ | $I_D$ | $G$-Band, cm$^{-1}$ | $I_G$ | $I_D/I_G$ | $L_a$, nm |
|---|---|---|---|---|---|---|
| GO | 1329 | 42 | 1585 | 95 | 0.44 | 87.5 |
| rGO | 1347 | 77 | 1593 | 99 | 0.78 | 49.4 |
| TiO$_2$-rGO (5%) | 1345 | 302 | 1605 | 381 | 0.79 | 48.7 |
| TiO$_2$-rGO (15%) | 1341 | 138 | 1601 | 154 | 0.90 | 42.8 |

The recalculated values of $L_a$ are shown in Table 1. The La values decreased from GO over rGO to the prepared nanocomposites. It could be concluded that prepared nanocomposites had decreased sp$^2$ domains in the structure [31].

XRD patterns of pure graphite and synthesized GO are given in Figure 3A. Both of these materials had characteristic X-ray diffraction peaks. The pure graphite showed characteristic peaks at 26.48° and 54.58° attributed to the (002) and (004) planes. In particular, the diffraction peak of GO located at 10.68° corresponded to the (001) plane. On the XRD pattern of rGO, after the reduction method of GO, a new and wide diffraction peak located at 24.40° attributed to the (002) plane appeared. The (002) plane in rGO indicates that the most oxygen-containing groups such as O–H, C=O, and C–O were removed from the nanomaterials. The new diffraction peak confirmed the successful conversion of GO into rGO (inset of Figure 3A) [44–46]. XRD patterns of pure TiO$_2$ and prepared nanocomposites

are shown in Figure 3B. The pure TiO$_2$ nanoparticles had diffraction peaks at 25.30°, 37.86°, 47.98°, 53.94°, 55.06°, 62.62°, 68.68°, 70.18°, and 75.12° and were indexed to the (101), (103), (004), (200), (105), (211), (204), (116), (220), and (215) plane, respectively [37]. Specified diffraction peaks correspond to the anatase-phase of TiO$_2$ nanoparticles. As can be seen in Figure 3B pure TiO$_2$ nanoparticles and both nanocomposites show only anatase-phase peaks in the X-ray diffraction patterns. This could be a consequence of the overlapping of the powerful intensity of the anatase-phase peak at 25.30° and the low diffraction peak of the rGO at 25.40° [46,47]. Nevertheless, we noticed that the diffraction intensities of anatase peaks decreased in both nanocomposites, in particular the diffraction peak at 25.30°. The reason for that can be in an increasing amount of GO with defects in carbon atoms.

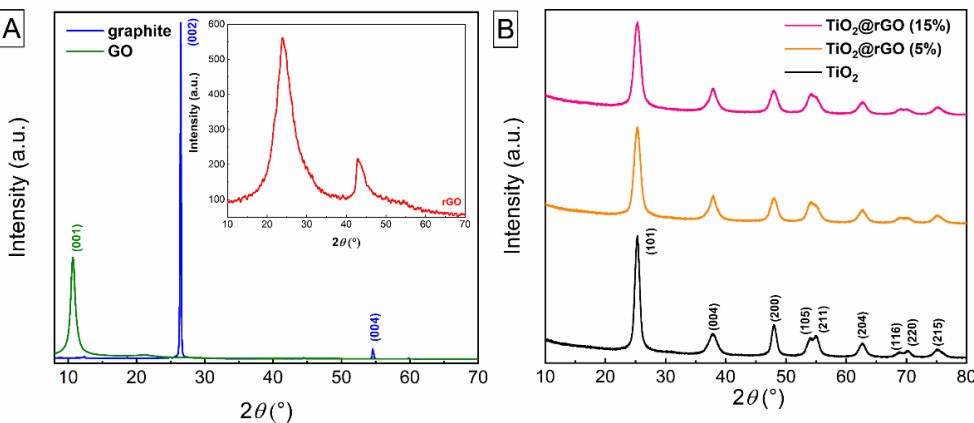

**Figure 3.** X-ray diffraction patterns. (**A**) pure graphite and synthesized GO (inset shows rGO); (**B**) TiO$_2$ and their synthesized nanocomposites.

XPS was performed in order to analyze the elemental composition and chemical environment of the elements detected at the surface of the prepared nanocomposite materials. Figure 4A shows a comparison between the high-resolution C 1s spectra of the different samples. The C 1s spectrum of GO shows the expected shape for this nanomaterial, with two intense peaks and a tail at higher BE. The peak at lower BE (284.4 eV) corresponded to the net of interconnected carbon atoms with a combination of sp$^2$ and sp$^3$ hybridization, derived from the breakdown of graphite layers [48]. The second peak and the tail were associated with carbon bonded to oxygen in a variety of functional groups, with a higher degree of oxidation as BE increased. This region of the spectrum was fitted using components centered at 286.4 eV, 288.2 eV, and 289.5 eV, corresponding to C–O, C=O, and O–C=O bonds, respectively [48]. In comparison, the rGO spectrum shows an intense peak at 284.4 eV with a smaller shoulder elongated towards higher BE values. The main peak, related to non-oxygenated carbon, was fitted using an asymmetric function according to the higher proportion of carbon sp$^2$ after the reduction of GO. The left side of the spectrum includes the contribution of different functional groups containing oxygen, just like in GO but with a noticeable decrease of intensity, indicating effective deoxygenation of GO nanosheets [48]. The carbon region of the TiO$_2$ sample reveals the typical spectrum shape of adventitious carbon contamination. It was fitted using the main peak at 285.3 eV, compatible with hydrocarbons, and two smaller components at 286.7 eV and 289.4 eV, related to the C–O and O–C=O groups. C 1s spectra of the TiO$_2$@rGO nanocomposites (5 wt.% and 15 wt.% of rGO) combine components from both rGO and TiO$_2$ samples. This fact was easily perceptible in the spectrum of TiO$_2$@rGO (5 wt.% of rGO), which showed a shape similar to the adventitious carbon spectra but with a shoulder at lower BE, towards the position of rGO. This way, both spectra (5 wt.% and 15 wt.% of rGO) were fitted using four components, the first one related to the C net of rGO, at 284.4 eV; the second one coinciding with the energy of hydrocarbons in TiO$_2$; and two more components associated to oxygen functionalities. As expected, there is a noticeable increase in the area of the component

associated with rGO in the nanocomposite containing 15 wt.% of rGO with respect to one having 5 wt.%.

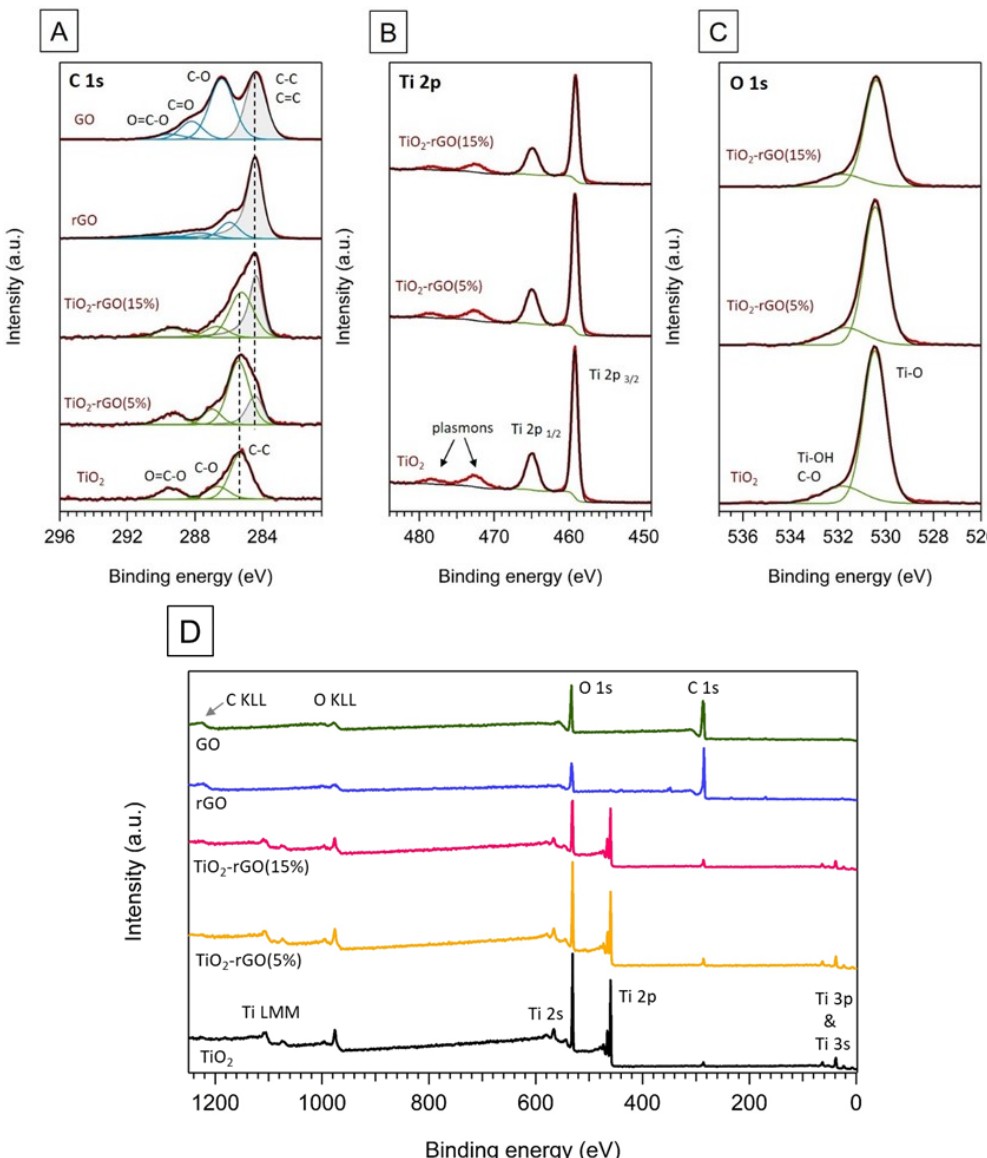

**Figure 4.** (**A**) C 1s, (**B**) Ti 2p, and (**C**) O 1s high-resolution XPS spectra of $TiO_2$ and $TiO_2$@rGO nanocomposites with different ratios prepared using the same procedure. In the case of C 1s (**A**), GO and rGO spectra were included for comparison and (**D**) full-spectrum scan.

High-resolution Ti 2p spectra of $TiO_2$ and $TiO_2$@rGO nanocomposites (Figure 4B) showed two symmetric peaks at 459.2 eV and 464.9 eV, corresponding to Ti $2p_{3/2}$ and Ti $2p_{1/2}$, respectively. Both the BE values and spin-orbit splitting of 5.7 eV were in good agreement with Ti (IV) in a $TiO_2$ chemical environment [44]. Supporting this, Figure 4C shows the O 1s spectra of $TiO_2$ and both $TiO_2$@rGO nanocomposites. All of them had a main peak at 530.4 eV, assigned to oxygen bonded to titanium, and a smaller and wider component around 531.8 eV, commonly ascribed to hydroxyl groups covering the surface oxygen vacancies in the $TiO_2$ structure [49,50], and also compatible with oxygen in organic compounds.

The synthesized $TiO_2$ and $TiO_2$@rGO nanocomposites and commercially available P-25 were analyzed by scanning electron microscope (SEM). Representative images of the analyzed powders are shown in Figure 5. The size of P-25 grains was around 30 nm (Figure 5A). The synthesized $TiO_2$ particles were half smaller and were spherical in shape, just like P-25 (Figure 5B). SEM images of synthesized GO and prepared rGO are displayed in Supplementary Figure S1. Figure 5C,D shows the $TiO_2$@rGO nanocomposites with 5 and 15 wt.% of rGO, which aggregated into larger forms. It can be seen that $TiO_2$ particles had a similar size as pure powder, and they were embedded in the surface of rGO sheets (insets in Figure 5C,D) [51,52]. To confirm the homogeneous distribution of elements in the prepared nanocomposite of $TiO_2$@rGO (15 wt%), elementary mapping is shown in Supplementary Figure S2.

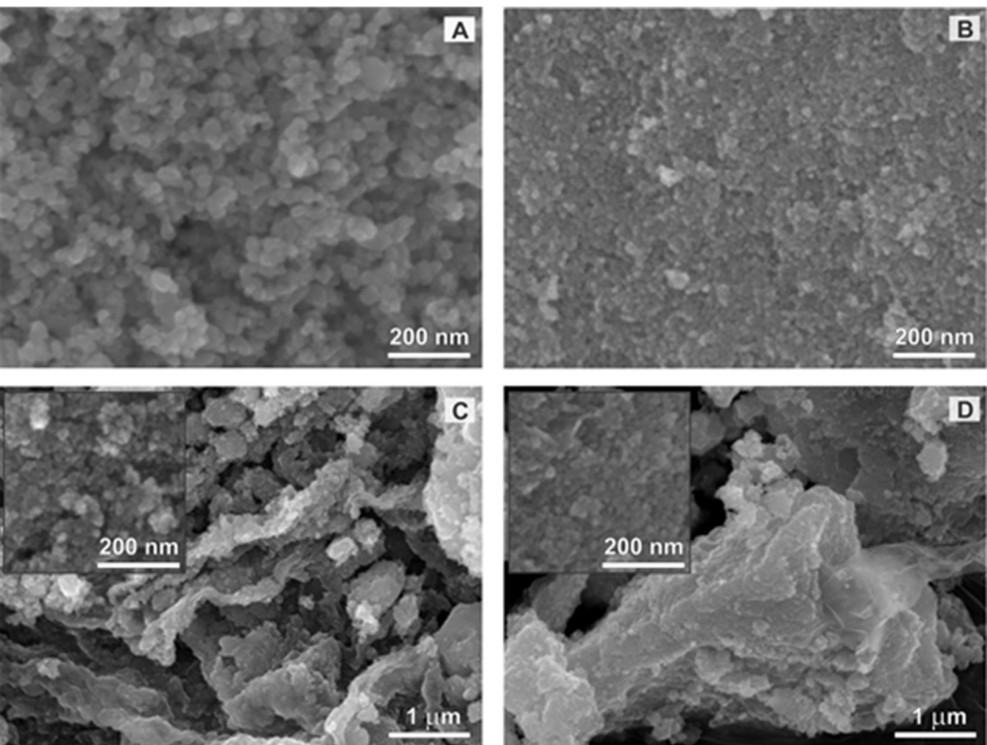

**Figure 5.** SEM images. (**A**) commercial P-25; (**B**) synthesized $TiO_2$; (**C**) $TiO_2$@rGO nanocomposite with 5 wt.% of rGO; (**D**) $TiO_2$@rGO nanocomposite with 15 wt.% of rGO.

### 3.2. Photocatalytic Activity of TiO₂@rGO Nanocomposites

Diffuse reflectance UV-Vis spectroscopy for modulation of the bandgap energy of the synthesized materials was calculated by Kubelka-Munk function, as shown in Figure 6. The bandgap results were obtained, giving $E_g$ values of 2.71 eV, 3.03 eV, and 3.15 eV for synthesized $TiO_2$@rGO nanocomposites and for pure $TiO_2$ particles. The bandgap energy of reference P-25 material was compared and displayed in the Supplementary Figure S3. The obtained results indicate that the decrease of the bandgap of the synthesized nanocomposites occurs with the introduction of rGO sheets. It can be observed that the photoactivity could be shifted into visible light.

Firstly, the photolysis tests (without catalysts) were performed, and it can be reported that the degradation of MB and RhB dyes was negligible under the irradiation. Before the photocatalytic tests, the adsorption properties of synthesized catalysts were evaluated. The obtained results of adsorption tests for MB and RhB dyes in the dark for different catalysts are displayed in Figure 7. The adsorption test is related to adsorption-desorption equilibrium among MB and RhB dyes molecules and catalysts surface. It could be seen that prepared nanocomposites showed higher adsorption of dyes compared to pure $TiO_2$ particles and reference P-25 material. Indeed, the dyes adsorption for pure $TiO_2$ particles

and reference P-25 material was negligible for both dyes. On the other hand, the RhB and MB adsorption for synthesized nanocomposites was dependent on the amount of rGO. The RhB adsorption for TiO$_2$@rGO (5%) was 1.6% which was lower than for the MB at 4.8%. The highest adsorption capacity of dyes was observed for TiO$_2$@rGO (15%) nanocomposite with 25.8% for RhB and 13.8% for MB.

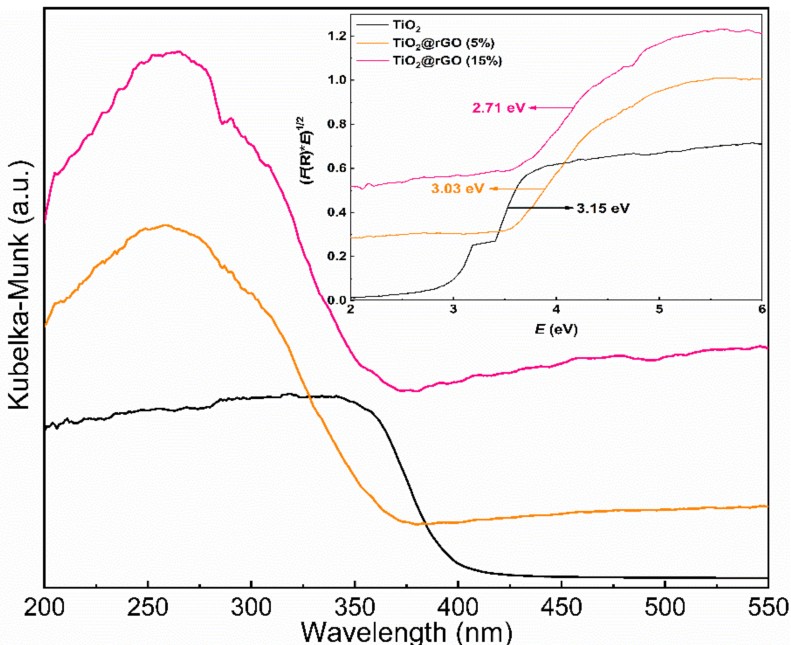

**Figure 6.** The absorption threshold energy (inset shows the bandgap energy) of synthesized pure TiO$_2$ particles and nanocomposites TiO$_2$@rGO with 5 wt% and 15 wt.% of rGO thermal treated at 300 °C.

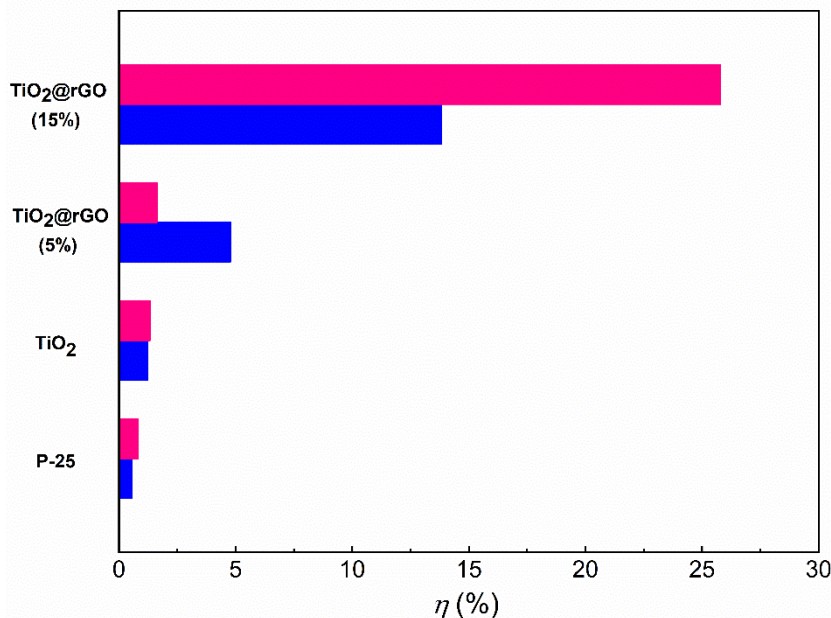

**Figure 7.** The percentage of adsorption for MB (blue) and RhB (pink) dyes after 60 min of stirring in the dark.

The photocatalytic performances of TiO$_2$ and respective nanocomposites were evaluated for dyes degradation under experimental parameters as reported in Supplementary Table S1. The photoactivity of prepared photocatalysts in comparison with reference P-25 material was monitored based on MB (cationic dye) and RhB (zwitterionic dye) decomposition under the irradiation (depicted in Figure 8A,B).

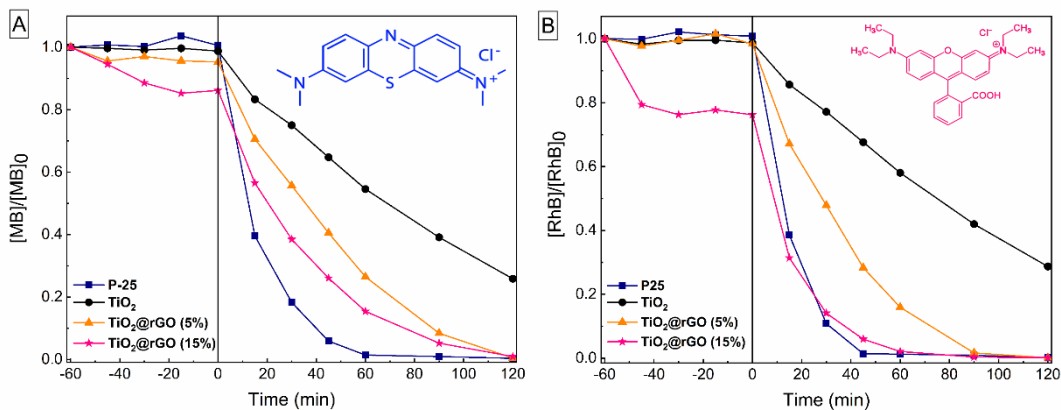

**Figure 8.** Photocatalytic degradation of (**A**) MB and (**B**) RhB in the dark (−60 min to 0 min) and under radiation (0 min to 120 min).

It can be seen in Figure 9A,B that the synthesized nanocomposites increased the decomposition of RhB and MB significantly compared to synthesized TiO$_2$ photocatalyst. Both synthesized nanocomposites have a very efficient MB and RhB decolorization after 120 min, which was very similar to reference P-25 material. The photoactivity of TiO$_2$@rGO (5%) was obtained with 99.6% of MB degradation and 99.9% of RhB photodecomposition. The nanocomposite with a higher amount of rGO (TiO$_2$@rGO (15%)) showed similar photodegradation results, 98.1% for (MB) and 99.8% for (RhB). Meanwhile, the photoactivity evaluation of synthesized TiO$_2$ particles was significantly lower than that of nanocomposites and referenced P-25 material. The degradation rate of TiO$_2$ was 70.9% for RhB and 73.9% for MB.

The photodegradation rate of MB and RhB dyes followed the pseudo-first-order kinetics. The linear kinetic rate constant (*k*) in detail is reported in Table 2. It could be determined that pseudo-first-order rate constants of TiO$_2$@rGO nanocomposites were significantly higher than for pure TiO$_2$ nanoparticles in the degradation of both dyes. The structure of organic pollutants played an important role in the photocatalytic process. MB as cationic dye and RhB as zwitterionic dye based on obtained photocatalytic tests showed significant photodegradation rates under the irradiation by nanocomposites which were related to the presence of rGO. In comparison to the P-25 TiO$_2$, for MB photodegradation, all composite materials showed lower photocatalytic activity, and for RhB, the photocatalytic activity was almost equal to the photoactivity of P-25 TiO$_2$. Pure synthesized TiO$_2$ showed the lowest photoactivity for both dyes. To confirm the immutability and persistence of as-prepared materials after performed photocatalytic tests, the SEM images were taken and given in Supplementary Figure S4.

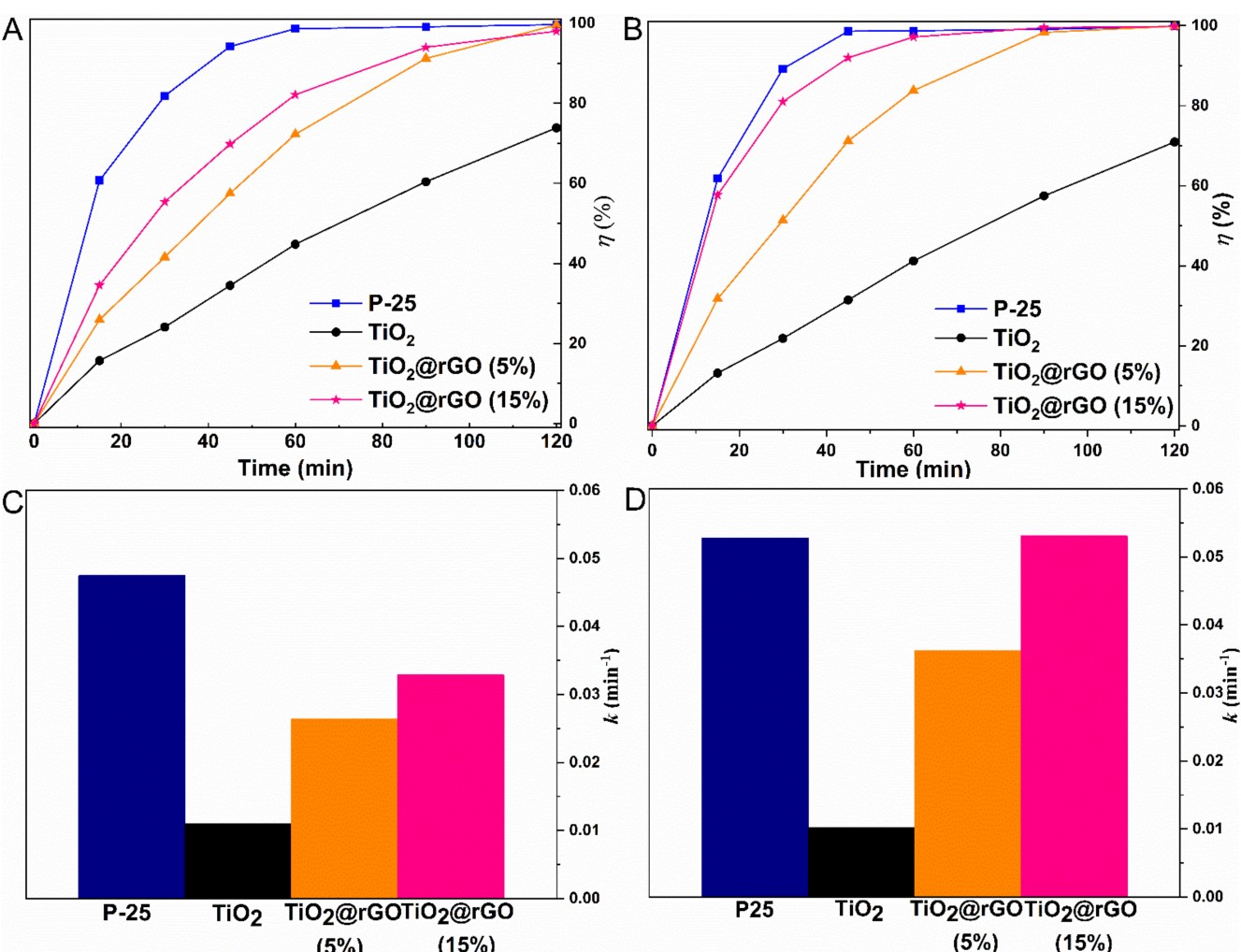

**Figure 9.** The percentage of (**A**) MB and (**B**) RhB degradation efficiency by the photocatalytic activity of the prepared catalysts under the radiation; rate constant, *k* of (**C**) MB and (**D**) RhB degradation for the as-prepared photocatalysts.

**Table 2.** Photodegradation kinetics of MB and RhB dyes under radiation.

| Sample ID | MB Dye | | | RhB Dye | | |
|---|---|---|---|---|---|---|
| | $R^2$ | $k \times 10^{-3}$, Min$^{-1}$ | $t_{1/2}$, Min | $R^2$ | $k \times 10^{-3}$, Min$^{-1}$ | $t_{1/2}$, Min |
| P-25 | 0.9919 | 47.45 | 14.61 | 0.9813 | 52.76 | 13.14 |
| TiO$_2$ | 0.9963 | 10.99 | 63.07 | 0.9958 | 10.19 | 68.02 |
| TiO$_2$@rGO (5 wt.%) | 0.9821 | 26.40 | 26.26 | 0.9688 | 36.15 | 19.17 |
| TiO$_2$@rGO (15 wt.%) | 0.9936 | 32.84 | 21.11 | 0.9951 | 53.05 | 13.07 |

## 4. Conclusions

The presented TiO$_2$@rGO nanocomposites were successfully synthesized with appropriate structural features for the possible application on the photodegradation of MB and RhD dyes under natural sunlight. Graphene-based TiO$_2$ nanocomposites, prepared using an optimized two-step method that combined hydrothermal treatment and calcination, presented a significantly improved photocatalytic activity under solar-like irradiation. The chemical bonding of particles was closely associated with the amount of rGO, time of hydrothermal synthesis, and calcination temperature. Importantly, the obtained value of the bandgap energy ($E_g$ = 2.71 eV and 3.03 eV) of nanocomposites indicate that the wavelength value shifted to visible light when compared with pure TiO$_2$ ($E_g \leq$ 3.15 eV). The application of prepared TiO$_2$@rGO nanocomposites as photocatalysts for degradation

of RhB and MB in an aqueous medium showed that the chemical integration of rGO with $TiO_2$ promoted synergistic effects which sped up the photodegradation of selected pollutants. The photodegradation efficiency of MB and RhB dye by the synthesized $TiO_2$@rGO (5 wt.% of rGO) and $TiO_2$@rGO (15 wt.% of rGO) nanocomposite photocatalysts after 120 min of exposure to the irradiation were significantly higher than for pure synthesized $TiO_2$ nanoparticles. $TiO_2$@rGO (5%) presented a photodegradation of 99.6% for MB and 99.9% for RhB. The nanocomposite with a higher amount of rGO ($TiO_2$@rGO (15%)) showed similar photodegradation results, 98.1% for MB and 99.8% for RhB. Meanwhile, the photoactivity evaluation of synthesized $TiO_2$ particles was significantly lower. Although these nanocomposite materials showed lower photocatalytic activity than P-25, their large layered structure allows a facile recovery from the aqueous medium after the photocatalytic reaction. These preliminary results show that the prepared $TiO_2$@rGO nanocomposite photocatalyst may be explored as high-efficiency and green photocatalysts for simultaneous sorption and degradation of dyes that can be directly applicable in real effluent water treatment by solar exposition.

**Supplementary Materials:** The following are available online at https://www.mdpi.com/article/10.3390/app11093966/s1.

**Author Contributions:** Conceptualization, M.K., M.P., and G.G.; methodology, M.K., L.Ć., D.L., and G.G.; software, M.K., G.O.-I., M.J.H., and G.G.; validation, M.K.; formal analysis, M.K., I.B. (Ivana Bačić), K.M., T.R., G.O.-I., M.J.H., M.P., and I.B. (Igor Bdikin); investigation, M.K.; resources, L.Ć., I.B. (Ivana Bačić), M.P., and G.G.; data curation, M.K., L.Ć., I.B. (Igor Bdikin), G.O.-I., M.J.H., and M.P.; writing—original draft preparation, M.K., I.B. (Ivana Bačić), G.O.-I., M.J.H., and M.P.; writing—review and editing, L.Ć., D.L., and G.G.; visualization, M.K. and G.G.; supervision, L.Ć. and G.G.; project administration, M.K., L.Ć., and G.G.; funding acquisition, L.Ć. All authors have read and agreed to the published version of the manuscript.

**Funding:** This work was supported by the projects UIDB/00481/2020 and UIDP/00481/2020—FCT—Fundação para a Ciencia e a Tecnologia; and CENTRO-01-0145-FEDER-022083—Centro Portugal Regional Operational Programme (Centro 2020), under the PORTUGAL 2020 Partnership Agreement, through the European Regional Development Fund. Thanks, are also due for the financial support to the $H_2O$ Value project (PTDC/NAN-MAT/30513/2017) also supported by FCT/MEC through national funds, and the co-funding by the FEDER, within the PT2020 Partnership Agreement and Compete 2020 (CENTRO-01-0145-FEDER-030513). The present study was also supported by the Slovenian Research Agency (ARRS) under the Contracts J2-9440 and L2-1830.

**Institutional Review Board Statement:** Not applicable.

**Informed Consent Statement:** Not applicable.

**Data Availability Statement:** The data presented in this study are available upon request from the corresponding author.

**Acknowledgments:** The authors are very thankful to the Centre for Mechanical Technology and Automation (TEMA), University of Aveiro, grant FCT: UID/EMS/00481/2019 and the grant FCT: IF/00582/2015 for helping in the characterization of materials. We thank the CENN Nanocenter for the use of Raman. Gil Gonçalves thanks to the Portuguese Science Foundation (FCT) for funding via the Programme: Stimulus of Scientific Employment—Individual Support (CEECIND/01913/2017).

**Conflicts of Interest:** The authors declare no conflict of interest.

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
