# Peer review of "Graphene-Based TiO2 Nanocomposite for Photocatalytic Degradation of Dyes in Aqueous Solution under Solar-Like Radiation"

_applsci, doi:10.3390/app11093966_

Round 1
Reviewer 1 Report
The authors have presented a novel method of photocatalytic degradation of dyes using Ti-O-C. The paper is well written. There are a few points that need to be revised before considering for publication.
- The authors may want to add the following references to further strengthen the introduction, where the use of TiO2 as a photocatalyst agent has been explained. DOI: 10.3390/ma13163511 and DOI: 10.1039/C9NH00590K
- The scale bar on SEM images are not very clear.
- Please assign the peaks in the Raman and XRD figures.
- could the author add the full XPS spectrum to the Figure.4
Author Response
The authors have presented a novel method of photocatalytic degradation of dyes using Ti-O-C. The paper is well written. There are a few points that need to be revised before considering for publication.
We thank the reviewer for the positive comments regarding the work reported on the manuscript.
The authors may want to add the following references to further strengthen the introduction, where the use of TiO2 as a photocatalyst agent has been explained.
The references suggested by the reviewer, DOI: 10.3390/ma13163511 and DOI: 10.1039/C9NH00590K, were added to the manuscript.
The scale bar on SEM images are not very clear.
The SEM images were changed, and a new scale bar was introduced.
Please assign the peaks in the figures.
The Raman and XRD peaks were assigned in the respective figures.
Could the author add the full XPS spectrum to the Figure.4
The full XPS spectrum was added to Figure 4 (D).

Reviewer 2 Report
In this work, "Graphene-based TiO2 nanocomposite for photocatalytic degradation of dyes in aqueous solution under solar-like radiation", the authors proposed a novel method for the development of TiO2/rGO nanocomposites for photocatalytic degradation of dyes in an aqueous solution. Based on the obtained results, the authors claimed that the potential of graphene-based TiO2 nanocomposite can be explored towards highly efficient solar light-driven photocatalyst for water purification. Overall, this manuscript has a strong potential for a second review after applying the issues and addressing the shortcomings listed below:
1-The authors should polish/revise some grammatical mistakes and typos along the manuscript. I invite the authors to read their manuscript carefully and make the required changes where necessary.
2-In the Introduction section, while discussing graphene-based materials and their possible application areas, the following works should be considered and cited to give a more general view to the possible readers of the work: [(i) Gated graphene enabled tunable charge-current configurations in hybrid plasmonic metamaterials, ACS Applied Electronic Materials 1, 637-641 (2019); (ii) Gated graphene island-enabled tunable charge transfer plasmon terahertz metamodulator, Nanoscale 11, 8091-8095 (2019)].
3-Corresponding references should be given if the considered equations are taken from some other study.
4-If possible, try to increase the resolution of each Figure.
5-In Figures 2 and 6, increase the font size of insets.
6-In Figures 9a and 9b, put the legends to the bottom right corner and increase the font size.
Author Response
In this work, "Graphene-based TiO2 nanocomposite for photocatalytic degradation of dyes in aqueous solution under solar-like radiation", the authors proposed a novel method for the development of TiO2/rGO nanocomposites for photocatalytic degradation of dyes in an aqueous solution. Based on the obtained results, the authors claimed that the potential of graphene-based TiO2 nanocomposite can be explored towards highly efficient solar light-driven photocatalyst for water purification. Overall, this manuscript has a strong potential for a second review after applying the issues and addressing the shortcomings listed below:
1-The authors should polish/revise some grammatical mistakes and typos along the manuscript. I invite the authors to read their manuscript carefully and make the required changes where necessary.
We thank the reviewer for the positive comments regarding the work reported on the manuscript. We carefully revised the manuscript to find some grammatical mistakes or typos.
2-In the Introduction section, while discussing graphene-based materials and their possible application areas, the following works should be considered and cited to give a more general view to the possible readers of the work: [(i) Gated graphene enabled tunable charge-current configurations in hybrid plasmonic metamaterials, ACS Applied Electronic Materials 1, 637-641 (2019); (ii) Gated graphene island-enabled tunable charge transfer plasmon terahertz metamodulator, Nanoscale 11, 8091-8095 (2019)].
We add the respective references to the introduction to give a broad overview of the application of GO to the readers.
3-Corresponding references should be given if the considered equations are taken from some other study.
Reference 33 was included in the main text of the manuscript related to equations used.
4-If possible, try to increase the resolution of each Figure.
We increased the resolution of the Figures.
5-In Figures 2 and 6, increase the font size of insets.
We increased the font size of the insets in Figures 2 and 6.
6-In Figures 9a and 9b, put the legends to the bottom right corner and increase the font size.
We add the legends to the bottom right corner of Figure 9a) and b) and increased the size of the letter.

Round 2
Reviewer 2 Report
In its current form, the revised manuscript is suitable for publication.